# Inducing Mitotic Catastrophe as a Therapeutic Approach to Improve Outcomes in Ewing Sarcoma

**DOI:** 10.3390/cancers15204911

**Published:** 2023-10-10

**Authors:** Soumya M. Turaga, Vikalp Vishwakarma, Stacey L. Hembruff, Benjamin K. Gibbs, Priya Sabu, Rajni V. Puri, Harsh B. Pathak, Glenson Samuel, Andrew K. Godwin

**Affiliations:** 1Department of Pathology and Laboratory Medicine, University of Kansas Medical Center, Kansas City, KS 66160, USA; sturaga@kumc.edu (S.M.T.); vvikalpksbt@gmail.com (V.V.); b942g123@kumc.edu (B.K.G.); rpuri@kumc.edu (R.V.P.); hpathak@kumc.edu (H.B.P.); 2University of Kansas Cancer Center, Kansas City, KS 66160, USA; shembruff@kumc.edu (S.L.H.); psabu@kumc.edu (P.S.); 3Division of Gynecologic Oncology, University of Kansas Medical Center, Kansas City, KS 66160, USA; 4Kansas Institute for Precision Medicine, University of Kansas Medical Center, Kansas City, KS 66160, USA; 5Division of Pediatric Hematology Oncology and Bone Marrow Transplantation, Children’s Mercy Hospital, Kansas City, MO 64108, USA; gsamuel1@cmh.edu; 6Division of Genomic Diagnostics, Department of Pathology and Laboratory Medicine, University of Kansas Medical Center, 3901 Rainbow Boulevard, MS 3040, Kansas City, KS 66160, USA

**Keywords:** Ewing sarcoma, drug synergy, kinesin family member 11, Aurora kinase A, SB-743921, VIC-1911

## Abstract

**Simple Summary:**

Ewing sarcoma (EWS) is a rare pediatric sarcoma affecting children and adolescents, with median diagnosis around the age of 15. Despite an intensive therapeutic regimen, including chemotherapy, surgery/radiation patients with recurrent (10–15%) and metastatic disease (<30%) have poor overall survival rates. Moreover, standard chemotherapy is intense and is often associated with systemic toxicity and secondary malignancies. Hence, it is critical to find new treatments to improve outcomes in EWS patients. We identified a combination of mitotic inhibitors targeting KIF11 (SB-743921) and AURKA (VIC-1911) that are effective in inhibiting EWS tumor growth at physiologically relevant nanomolar doses. This drug combination inhibited EWS cell viability in vitro by promoting cell cycle arrest followed by cell death. In vivo, this treatment regimen led to significantly delayed tumor growth and improved overall survival in xenograft EWS mouse models. Overall, these preclinical data provide encouragement to consider a future clinical trial for patients with this deadly disease.

**Abstract:**

Ewing sarcoma (EWS) is an aggressive pediatric malignancy of the bone and soft tissues in need of novel therapeutic options. To identify potential therapeutic targets, we focused on essential biological pathways that are upregulated by EWS-FLI1, the primary oncogenic driver of EWS, including mitotic proteins such as Aurora kinase A (AURKA) and kinesin family member 15 (KIF15) and its binding partner, targeting protein for Xklp2 (TPX2). KIF15/TPX2 cooperates with KIF11, a key mitotic kinesin essential for mitotic spindle orientation. Given the lack of clinical-grade KIF15/TPX2 inhibitors, we chose to target KIF11 (using SB-743921) in combination with AURKA (using VIC-1911) given that phosphorylation of KIF15^S1169^ by Aurora A is required for its targeting to the spindle. In vitro, the drug combination demonstrated strong synergy (Bliss score ≥ 10) at nanomolar doses. Colony formation assay revealed significant reduction in plating efficiency (1–3%) and increased percentage accumulation of cells in the G2/M phase with the combination treatment (45–52%) upon cell cycle analysis, indicating mitotic arrest. In vivo studies in EWS xenograft mouse models showed significant tumor reduction and overall effectiveness: drug combination vs. vehicle control (*p* ≤ 0.01), SB-743921 (*p* ≤ 0.01) and VIC-1911 (*p* ≤ 0.05). Kaplan–Meier curves demonstrated superior overall survival with the combination compared to vehicle or monotherapy arms (*p* ≤ 0.0001).

## 1. Introduction

Ewing sarcoma (EWS) is the second most common pediatric bone sarcoma after osteosarcoma affecting kids and young adults. EWS is a rare cancer with an annual incidence of 2.93 cases per million per year [1]. The pathogenic event in EWS is a somatic chromosomal translocation resulting in a fusion oncogene, *EWS-ETS,* and efforts to target it therapeutically have been unsuccessful. Chemotherapy has improved the 5-year overall survival rate for patients with localized disease (to approximately 60–70%) but unfortunately is less effective on metastatic (30%), refractory, and recurrent disease (<15%), indicating that there is still a great unmet need for effective therapies [2]. To identify new therapies for EWS, we previously conducted a multi-pronged approach using in silico predictions of drug activity via an integrated bioinformatics approach in parallel with an in vitro screen of FDA-approved drugs. We uncovered drug targets [3,4] essential for mitotic spindle formation and cell cycle progression such as kinesin family member 15 (KIF15) and its binding partner TPX2 [5] and Aurora kinase A (AURKA), which are upregulated by *EWS-FLI1*, which is the most predominant EWS-ETS fusion and the primary oncogenic driver of EWS.

KIF15 is a plus-end directed kinesin that localizes to spindle microtubules and chromosomes and plays a role in maintaining spindle bipolarity during mitosis [6]. Though KIF15 is not essential for bipolar spindle formation during normal cell division, KIF15 compensates when the function of key motor kinesin family member 11 (KIF11/Eg5) is inhibited [7]. There are several KIF11 inhibitors that can disrupt the mitotic spindle bipolarity function mediated by KIF11; however, in most studies, resistance is observed as KIF15 replaces the functions of KIF11 in a TPX-2-dependent manner [7,8]. Importantly, we and others have shown that KIF15 cooperates with KIF11 to promote bipolar spindle assembly and formation [9], which is essential for proper sister chromatid segregation, and when KIF15 is genetically silenced, the efficacy of KIF11 inhibition is significantly enhanced. In previous studies, we designed an RNAi-based screen of the “druggable genome” to identify putative points of molecular vulnerability across a diverse panel of ovarian cancer cell lines [10]. These screens identified KIF11 as an essential protein in maintaining tumor cell viability. KIF11 is a tetrameric crosslinker and mitotic motor protein that facilitates mitotic progression through metaphase and anaphase by binding and pushing apart microtubules in the bipolar spindle. Hence, dual inhibition of KIF11 and the KIF15/TPX2 axis is essential to disrupt the mitotic activity of cancer cells.

AURKA is a serine/threonine kinase with crucial functions in mitosis and has aberrant expression in most cancer types [11,12,13]. AURKA-mediated phosphorylation regulates the functions of a diverse set of AURKA substrates, some of which are mitosis regulators, including KIF15 [14]. Studies have shown that dual inhibition of KIF11 and AURKA can overcome KIF15-dependent drug resistance, and in KIF11 inhibitor-resistant HeLa cells, dual inhibition of KIF11 and AURKA led to the formation of monopolar spindles, indicating the potency of combined targeting of these proteins [15]. Due to the lack of clinically relevant inhibitors to target KIF15 directly, we sought to indirectly target KIF15 using a new clinically relevant AURKA inhibitor, VIC-1911. Previous studies have shown that AURKA directly regulates KIF15 and that phosphorylation of KIF15^S1169^ by Aurora A is required for its targeting to the spindle [14]. VIC-1911 (developed by VITRAC Therapeutics, LLC), formerly known as TAS-119, is a novel, selective, and orally active small molecule inhibitor of AURKA developed for the treatment of solid tumors and hematologic malignancies. In preclinical studies, VIC-1911 demonstrated anti-tumor activity [16,17] and is currently being studied in phase 1 clinical trials for advanced solid tumors [18].

KIF11 has also been identified by the pharmaceutical industry as a viable target to develop anti-cancer drugs [19,20,21]. Although these KIF11 inhibitors are generally well tolerated by patients [22], the clinical response rates as monotherapies in adult patients with advanced cancers are typically less than 10% [23,24]. SB-743921 (also known as kinesin spindle protein inhibitor) is a second-generation small molecule ATPase inhibitor of KIF11. It has been reported to have greater than 40,000-fold sensitivity for KIF11 over other kinesins [25]. SB-743921 has been used in several preclinical studies, where it has demonstrated significant anti-tumor activity, and it is being evaluated in clinical trials for multiple cancers [25,26]. Both KIF11 and AURKA inhibitors have been employed in clinical trials as single agents or in combination with chemotherapy but have not been efficacious, indicating the need to reevaluate their mode of action and clinical limitations [12,27,28]. Based on these clinical outcomes, we sought to evaluate these single-drug agents by the dual targeting of key mitotic regulators. To our knowledge, the following is the first study to report the in vitro synergistic activities and in vivo anti-tumor efficacy of this drug combination in EWS.

## 2. Materials and Methods

### 2.1. DepMap Portal Gene Expression Analysis

RNA expression levels of *KIF11*, *KIF15*, *AURKA*, and *TPX2* across the Cancer Cell Line Encyclopedia were obtained from the DepMap Portal (RRID:SCR_017655) published by The Broad Institute [29]. Expression levels were separated by cancer type and sorted by median value. Cancer types with two or fewer representatives or of hematological origin were excluded from analysis. This yielded 903 samples across 26 different tumor types. Data were plotted using GraphPad Prism 9.5 software (RRID:SCR_002798).

### 2.2. Cell Lines and Cell Culture

We selected a panel of confirmed cell lines that are representative of the most common *EWS-ETS* fusion types, namely, *EWS-FLI1* type I (CHLA-9, CHLA-10, CHLA-32, TC-32, and TC-71), *EWS-FLI1* type II (SK-ES-1), *EWS-FLI1* type III (CHLA-258) fusion, and *EWS-ERG* fusion (COG-E-352) [30]. For non-EWS controls, rhabdomyosarcoma cell line RMS-13, osteosarcoma cell line U2-OS, and bone marrow-derived mesenchymal stem cells (BMD-MSCs) were utilized. BMD-MSCs (ATCC #PCS-500-012, (RRID:CVCL_A0YN), SK-ES-1 (ATCC #HTB-86, RRID:CVCL_0627), U2-OS (ATCC # HTB-96, RRID:CVCL_0042), and RMS-13 (ATCC #CRL-2061,RRID:CVCL_0041) were purchased from the American Type Culture Collection (ATCC). In addition, TC-71 (RRID: CVCL_2213), TC-32 (RRID: CVCL_7151), CHLA-32 (RRID: CVCL_M151), CHLA-9 (RRID:CVCL_M150), CHLA-10 (RRID:CVCL_6583), COG-E-352 (RRID:CVCL_M153), and CHLA-258 (RRID:CVCL_A058) cell lines were obtained from the Children’s Oncology Group (COG). All cell lines mentioned above have been authenticated by STR profiling. TC-71, CHLA-9, CHLA-10, CHLA-32, CHLA-258, and COG-E-352 were cultured in Iscove’s Modified Dulbecco’s Medium (Gibco #12440-053) supplemented with 20% FBS, 2 mM L-glutamine, and 1X Insulin-Transferrin-Selenium (Gibco #41400-045). SK-ES-1 was cultured in McCoy’s 5A medium (ATCC #30-2007) with 15% FBS, and TC-32, CRL-2061, and U2-OS were cultured in RPMI-1640 (HyClone #SH30027.01) medium with 10% FBS. BMD MSC cells were cultured in MSC basal media (ATCC# PCS-500-030) supplemented with 7% FBS, rhIGF-1 (15 ng/mL, R&D systems #233-FB-025), rhFGF-b (125 pg/mL, R&D systems #291-G1-200), and L-alanyl-L-glutamine (2.4 mM, Fisher scientific # AAJ6699606).

### 2.3. Drugs

Ispinesib (MedChem Express, HY-50759), filanesib (MedChem Express, #HY-15187), SB-743921 (MedChem Express, #HY-12069), and VIC-1911 were obtained from VITRAC Therapeutics. The drugs were re-suspended in DMSO for in vitro studies.

### 2.4. Cell Viability and Drug Synergy Assay

In total, 500 cells/well (80 µL) were plated in triplicate in 96-well plates in an 8 × 8 matrix. Drugs (20 µL) were added the next day (18–24 h) after seeding the cells, and the cells were incubated for 72 h before cell viability readings were measured. CellTiter-Glo reagent (Promega # G7572) mixed with Glo Lysis buffer (Promega # E2661) in a 1:2 ratio was used for analysis. An equal volume of reagent to cell culture media was added. After adding the reagent, the plates were mixed and incubated for 10 min and the readings were measured using a TECAN Infinite 200 PRO plate reader (RRID:SCR_019033). The percentage inhibition of cell viability was calculated and the synergy scores were assessed using SynergyFinder (RRID:SCR_019318) [31]. Dose–response curves, heat maps, and 2D synergy plots were generated by the software, and the Bliss scoring algorithm was used to calculate the synergy. Synergy scores less than −10 indicate that the interaction between the two drugs was likely to have been antagonistic, scores from −10 to 10 indicate an additive effect, and scores higher than 10 indicate a synergistic effect.

### 2.5. Colony Formation Assay

In total, 500 cells/well (TC-71 and SK-ES-1) were plated in triplicate in a 6-well plate. The cells were treated with drugs the following day, after which the plates were incubated and observed for 7 days. After the 7-day period, the cells were washed twice with 2 mL of 1X PBS and fixed with 500 µL methanol for 30 min. The cells were then stained with 1 mL of crystal violet solution (0.1% *w*/*v*, Sigma Aldrich, Cat# C0775-25G) at room temperature for 30–40 min. The stained colonies were imaged and quantified using a Celigo Imaging Cytometer (Nexcelom Biosciences, RRID:SCR_018808). Data acquisition and imaging were completed in a blinded manner. The plating efficiency was calculated by the following formula: number of colonies formed/number of colonies plated × 100%.

### 2.6. Cell Cycle Analysis

TC-71 and CHLA-10 cells (1.5 × 10^5^ cells per well) were plated in triplicate in a 6-well plate for each condition. Drugs were added the next day and the cells were incubated for 24 h. After drug treatment, the cells were centrifuged at 300× *g* for 5 min and washed twice with 1× PBS solution. The cells were then fixed with 70% ethanol and stained with 500 µL of FxCycle™ PI/RNase Staining Solution (Thermo Fisher Scientific, Waltham, MA, USA, Cat# F10797). The DNA content was measured using an Attune™ NxT flow cytometer (Invitrogen, RRID:SCR_019590). The cells were gated based on the vehicle treatment of each cell line. Data were analyzed using FlowJo software v10 (BD Biosciences, RRID:SCR_008520). Data acquisition was completed in a blinded manner.

### 2.7. Capillary Western Blot (Wes) Analysis

Capillary Western analyses were performed using the ProteinSimple Wes System. Final concentration of 0.4 µg/µL of protein samples were used for analysis, and the samples were diluted with 0.1 × sample buffer. Then, 4 parts of the diluted sample were combined with 1 part 5x Fluorescent Master Mix (containing 5x sample buffer, 5x fluorescent standard, and 200 mM DTT) and heated at 95 °C for 5 min. The Fluorescent Master Mix contains three fluorescent proteins that act as a “ruler” to normalize the distance for each capillary because the molecular weight ladder is only on the first capillary and each capillary is independent. After this denaturation step, the prepared samples; blocking reagent; 1:50 diluted primary antibodies KIF11 or Eg5 (Cell Signaling Technology, Danvers, MA, USA, Cat# 4203, RRID:AB_10545760), p-KIF11 (Thermo Fisher Scientific Cat# PA5-105186, RRID:AB_2816659), KIF15 (Proteintech, Rosemont, IL, USA, Cat# 55407-1-AP, RRID:AB_11182836), AURKA (Novus Cat# NBP1-51843SS, RRID:AB_11018019), p-AURKA (Novus Cat# NBP3-05434) PARP(46D11) (Cell Signaling Technology Cat# 9532, RRID:AB_659884), and β-actin (Cell Signaling Technology Cat# 12262, RRID:AB_2566811); HRP-conjugated secondary antibodies anti-rabbit (Biotechne, Minneapolis, MN, USA, #DM-001) and anti-mouse (Biotechne, #DM-002); and chemiluminescent substrate were dispensed into designated wells in an assay plate. A biotinylated ladder provided molecular weight standards for each assay. After plate loading, the separation electrophoresis and immunodetection steps took place in the fully automated capillary system. The compass software 5.0 for Simple Western (RRID:SCR_022930) was used to analyze the data and process the results. For the quantification of the Wes blots, the area of the bands was used and was determined by the integrated analysis tool in the Wes Compass software 5.0. The lane normalization factor was calculated using the following formula: observed signal of housekeeping protein (β-Actin) for each lane/highest observed signal of housekeeping protein on the blot. The protein expression or normalized experimental signal for each protein/sample was calculated as observed experimental signal/lane normalization factor.

### 2.8. Tumor Xenograft and Drug Treatment Studies

Six-week-old female NSG mice were inoculated subcutaneously in the right flanks with a 200 µL suspension of TC-71 (2 × 10^6^ cells/site) mixed with an equal volume of ice-cold Matrigel (Corning, Corning, NY, USA, Cat#354234). An appropriate amount of SB-743921 drug was re-suspended in 10% DMSO, 40% PEG300, 5% Tween-80, and 45% PBS. VIC-1911 drug was re-suspended in 0.5% hydroxypropyl methylcellulose solution. After the tumors reached approximately 500 mm^3^, the mice were randomized into four treatment groups (*n* = 10 mice per group as per power analysis) and treated as follows: For study 1, the groups included (1) vehicle control, with an equivalent dose of SB-743921 vehicle (intraperitoneal) and VIC-1911 vehicle (oral gavage); (2) SB-743921 only (2.5 mg/kg); (3) VIC-1911 only (37.5 mg/kg); and (4) a combination of SB-743921 (2.5 mg/kg) and VIC-1911(37.5 mg/kg). The mice were treated with the vehicle and drugs every other day for 20 days. Study 2 included (1) vehicle control, with an equivalent dose of SB-743921 vehicle (intraperitoneal) and VIC-1911 vehicle (oral gavage); (2) SB-743921 only (1.25 mg/kg); (3) VIC-1911 only (37.5 mg/kg); and (4) a combination of SB-743921 (1.25 mg/kg) and VIC-1911(37.5 mg/kg). The mice were treated with the vehicle and drugs 3 times a week for 42 days. The tumor volume and body weight were measured 3 times per week. The tumor volumes were measured with calipers and calculated using the following formula: volume (mm^3^) = length × (width)^2^ / 2. The mice were humanely euthanized, and gross necropsies were performed when the tumor volumes exceeded 4000 mm^3^. The researchers were not blinded during drug treatment, data collection, or analysis.

### 2.9. Data Analysis and Statistics

In vitro data were reported as mean ± SD of 3 independent experiments. In the statistical analyses for the in vivo xenograft study, Kaplan–Meier survival curves were used to determine the difference in survival among the treatment groups. A *p*-value of less than 0.05 was considered statistically significant. All statistical analyses were performed using GraphPad Prism 9.5 software (RRID:SCR_002798).

## 3. Results

### 3.1. In Silico Bioinformatics Screen Identifies Mitotic Proteins Essential for EWS Growth

Using integrated bioinformatics and high-throughput screening, we previously identified and validated several drugs that were predicted to reverse the EWS disease signatures and/or EWS-ETS-dependent signatures [3]. We then computed and sorted the expression changes of individual genes associated with the EWS signatures upon treatment with the validated drugs. We identified the top 15 genes (Table 1) whose transcript levels were significantly (*p* < 0.05) downregulated by these drugs. Among these genes, KIF15 (encoding for a motor kinesin) was ranked 1st, and its potential binding partner, TPX2 (encoding for a microtubule-associated protein that mediates bipolar spindle assembly and mitosis), was ranked 14th on the list. Notably, in separate but parallel investigations conducted by our team, we observed significant vulnerability of the spindle assembly motor proteins KIF11 and KIF15 and their binding partner TPX2 in epithelial ovarian cancer (unpublished data). Furthermore, we found that elevated levels of KIF15 and TPX2 contributed to resistance against the KIF11 inhibitor (KIF11i), as demonstrated in both in vitro and in vivo studies. We also identified another key mitotic kinase, Aurora kinase A (AURKA), to be ranked sixth in our screen. Furthermore, bioinformatic analysis of RNA expression data from solid cancers within DepMap portal indicated that KIF15 expression levels were the highest in Ewing sarcoma and that KIF11 expression levels were the second highest in Ewing sarcoma, only behind synovial sarcomas (Figure 1A). Interestingly, our analyses found that both KIF11 and KIF15 were highly expressed across all the various sarcomas included in the portal.

STRING protein–protein interaction analysis of the top 15 candidates indicated an association between the proteins involved in cell cycle progression (Appendix A). Gene enrichment analysis indicated biological pathways such as AURKA signaling (Appendix A) and biological processes such as spindle assembly (Appendix A) to be the key pathways mediated by the top 15 genes. We next investigated publicly available databases to assess the RNA expression of transcripts encoding for KIF11, KIF15, TPX2, and AURKA in EWS patient tumor samples using BioGPS (E-GEOD-12102) [32] (Appendix A). These genes were expressed in all EWS patients, indicating the feasibility of targeting them. KIF11, KIF15, TPX2, and AURKA protein levels were abundantly elevated in several cell line models of EWS (Figure 1B,C, Appendix A). Given their important role in cell cycle progression, the bulk gene expression data from The Genotype-Tissue Expression database [33] suggests that their corresponding proteins are also present in normal tissues as well but at lower levels (Appendix A). Small molecule inhibitors to KIF15 such as KIF15-IN-1 have been tested in preclinical studies [34,35] but none of them have been advanced to clinical trials; hence, we chose to target the mitotic protein upstream of KIF15, i.e., AURKA, with clinically available inhibitors.

### 3.2. Synergistic Inhibition of EWS Growth In Vitro by KIF11 and AURKA Inhibitors

We performed drug synergy assays to identify the most synergistic KIF11 inhibitor in combination with AURKA inhibitor VIC-1911. Given that there are several KIF11 inhibitors that have been tested in clinical trials for different types of cancers, we tested the three most widely used inhibitors, e.g., ispinesib, filanesib, and SB-743921 (also known as kinesin spindle protein inhibitor). We initially tested these drug combinations in TC-71, an EWS-FLI1 Type I fusion-containing cell line. An 8 × 8 matrix was used to test different combinations of both inhibitors. The cell inhibition was measured using the CellTiter-Glo viability assay, and the drug synergy was calculated via Synergy Finder [31]. The Bliss algorithm was used to measure the synergy, based on which scores greater than 10 indicated significant synergistic interaction between the two drugs. We observed synergistic interaction with the three different KIF11 inhibitors used. The synergy scores were observed to be the highest with the SB-743921 and VIC-1911 combination (24.62) (Figure 2A) compared to the synergy scores of ispinesib (19.54) (Figure 2B) and filanesib (16.81) (Figure 2C) in combination with VIC-1911. Hence, we decided to use the combination of SB-743921+ VIC-1911 for this study.

### 3.3. Drug Synergy Is Observed in Different EWS Fusion Type Cell Lines

We next assessed whether the drug combination was sensitive to different EWS-ETS fusion types and tested the combination of SB-743921 and VIC-1911 across multiple EWS cell lines bearing different EWS fusions. EWS-FLI1 Type II (SKES-1) (Figure 3A), EWS-FLI1 Type III (CHLA-258) (Figure 3B), and EWS-ERG (COG-E-352) (Figure 3C) cell lines yielded strong synergistic drug activity with the combination treatment. This finding indicates that this drug combination is efficient at reducing EWS cell viability irrespective of fusion type. We also tested treatment naïve EWS-FLI1 Type I fusion-bearing cell lines CHLA-9, CHLA-32, and TC-32 with the combination (Appendix A) and found that these cells had greater susceptibility and synergy compared to the other cells lines established post-treatment. Again, these results suggest that dual inhibition is effective irrespective of the treatment status of EWS cells. Synergy was also observed in other cancer types, such as osteosarcoma (U2OS) and rhabdomyosarcoma (RMS-13) (Appendix A). However, in control BMD-MSCs (Appendix A), the presumptive progenitor cells to EWS [36], the combination was not effective in inhibiting cell growth, indicating enhanced sensitivity of these drugs to more rapidly proliferating cancer cells and a possible therapeutic window for this drug combination.

### 3.4. Combination Treatment with SB-743921 and VIC-1911 Reduces Colony Formation In Vitro

We next performed colony formation assay to measure the clonogenicity of the drug combination. Different drug combinations were tested alone and in combination. We used the drug dosages that were in the synergistic range and assessed the ability of single cells to form colonies. We seeded very low numbers (500 cells/well) of SKES-1 cells and TC-71 cells in a six-well plate, and drugs were added the next day after seeding. The cells were incubated for 7 days and were fixed and stained using crystal violet. The plates were imaged, and the colonies were quantified using a Celigo Imaging Cytometer. We observed significantly fewer colonies in the wells treated with the drug combination compared to single-drug treatment and the control group (Figure 4A) and a significant reduction in plating efficiency with the combination treatment. Mean plating efficiencies for SK-ES-1 cells were control (100.5%), 0.625 nM SB-743921 (103.3%), 0.312 nM SB-743921 (104.9%), 25 nM VIC-1911 (59.6%), 12.5 nM VIC-1911 (84%), 0.625 nM SB-743921 + 12.5 nM VIC-1911 (2%), and 0.625 nM SB-743921 + 25 nM VIC-1911 (0.8%). For TC-71 cells, they were control (113.1%), 0.625 nM SB-743921 (20%), 0.312 nM SB-743921 (48.9%), 25 nM VIC-1911 (60.7%), 12.5 nM VIC-1911 (38.6%), 0.625 nM SB-743921 + 12.5 nM VIC-1911 (1.6%), and 0.625 nM SB-743921 + 25 nM VIC-1911 (1.8%). (Figure 4B). These results indicate that the combination treatment had a significant effect on EWS cancer cell survival in vitro.

### 3.5. Cell Cycle Analysis Indicates Combination Treatment Arrests the Cells in G2/M Phase

KIF11 [37] and AURKA [38] inhibitors have known roles in mediating G2/M arrest and, in turn, halting cell division. We performed cell cycle analyses on TC-71 and CHLA-10 cells to test whether the drug combination was effective at arresting the cells in the G2/M phase compared to single drugs alone. We observed that there was an increase in the number of cells in the G2/M and sub G1 phases (cell fragments), indicating cell cycle arrest and apoptosis, respectively, with the combination treatment (Figure 5A). Quantification of the percentage of cells in the G2/M and subG1 phases indicated significant effects on cell growth and viability with the combination treatment (Figure 5B).

### 3.6. Protein Expression Post-Combination Treatment Indicates an Increase in Expression of KIF11 and AURKA

We performed a capillary electrophoresis-based Western blot assay (Wes) to determine protein expression levels following drug treatment. We first checked the protein levels of KIF11 and AURKA upon treatment with their specific inhibitors’ SB-743921 and VIC-1911. We observed that, compared to control and single-drug treatment groups, there was an increase in accumulation of KIF11 and AURKA proteins in the combination treatment group (Figure 6A,B, Appendix A). This finding could be due to increased accumulation of cells in the G2/M phase, as observed previously (Figure 5B). We next tested the status of AURKA and KIF11 phosphorylation following drug treatment and found enhanced phosphorylation of AURKA at Thr^288^ and of KIF11 at Thr^926^ and a corresponding decrease in KIF15 protein levels with the drug combination group. We also observed an increase in expression of cleaved-poly (ADP-ribose) polymerase (c-PARP) upon SB-743921 drug treatment, which was enhanced in combination with VIC-1911, indicating induction of apoptosis (Figure 6A,B, Appendix A). These finding continue to support drug targeting of KIF11 and AURKA and subsequent downregulation of KIF15 as a potential new therapeutic approach in children and young adults diagnosed with recurrent EWS. Taken together, these protein expression data indicate that combination treatment inhibits EWS growth via perturbation of the KIF11/15 pathways, contributing to G2/M arrest and apoptosis.

### 3.7. Combination Treatment Synergistically Leads to Tumor Regression in EWS Xenograft Mouse Model

To test the efficacy of the drug combination in vivo, we employed an EWS mouse xenograft model. Female NSG mice (6–8 weeks) were implanted subcutaneously with 2 × 10^6^ TC-71 cells. Once the tumors reached approximately 500 mm^3^, the mice were grouped into four treatment arms. In efficacy study 1, the mice were grouped into control, SB-743921 (2.5 mg/kg), VIC-1911 (37.5 mg/kg), or a combination (Appendix A) and were treated every other day for 20 days. We observed significant tumor regression with the drug combination compared to control or the VIC-1911 monotherapy arm (Appendix A). At day 32 (12 days after ending treatment), the animals treated with the drug combination demonstrated no evidence of measurable disease based on palpation. We followed survival (based on regulations by our IACUC) and observed that the tumor-bearing animals treated with the combination survived significantly longer (70 days) than the vehicle and monotherapy treatment arms (Appendix A). Despite the desired and significant efficacy observed, the animals displayed a significant loss in body weight when treated with 2.5 mg/kg of SB-743921 alone or a combination (Appendix A). Although the animals’ body weights recovered when treatment was discontinued, we designed a second efficacy study in which the dosage of SB-743921 was reduced to 1.25 mg/kg/treatment and the treatment regimen was altered. In efficacy study 2, tumor-bearing animals were treated with vehicle (control), SB-743921 (1.25 mg/kg), VIC-1911(37.5 mg/kg), or a combination three times a week (Monday, Wednesday, and Friday) for up to 6 weeks (42 days). The mice were monitored until defined end-point symptoms were observed (Figure 7A). Based on the approved IACUC protocol, the animals were euthanized when the tumor reached 4000 mm^3^. As shown in Figure 7B, the combination group continued to show the greatest level of efficacy compared to the vehicle control or the monotherapy arms. However, when using a lower dose of SB-743921 in combination with VIC-1911, the tumors recurred more quickly. The combination treatment also had a significant effect on improving the overall survival time of the mice compared to the control and single-drug treatment groups (Figure 7C). Compared to the animals in efficacy study 1, their body weights were more stable throughout the treatments (Appendix A). Combined, these data indicate that synergistic targeting of KIF11 and AURKA with specific inhibitors causes a significant delay in tumor growth and improved overall survival compared to single drugs alone.

## 4. Discussion

Antimitotic drugs are among the most important chemotherapeutic agents available to oncologists and continue to be a clinical staple in the treatment of most solid tumors, including EWS [39,40]. Newly diagnosed EWS patients are treated with an aggressive chemotherapeutic regimen that consists of 14 cycles with vincristine, doxorubicin, and cyclophosphamide, alternating with ifosfamide and etoposide (VDC/IE) [41]. These drugs have both short-term and long-term adverse effects in patients, such as myelosuppression, cardiotoxicity, neuropathy, and secondary malignancies [42,43]. Despite extensive therapy utilized commonly in new diagnoses, at least one-fourth of patients with localized disease will relapse after completion of therapy. Meanwhile, for newly diagnosed patients with metastatic disease, recurrence rates are even higher, with treatment failure seen in 50–80%, depending on the location of the metastases [44]. For patients with relapse or for those who are refractory to initial therapy, the odds of long-term survival are low. In addition, there is currently no standard management for this group of patients, raising many questions about how best to proceed. Hence, there is an unmet need for novel therapies, especially targeted agents for EWS for this patient population. Therefore, more specific inhibitors of mitosis could avoid the side effects of anti-microtubule agents (e.g., vincristine) such as neuropathy [45].

Current therapies for EWS that are under development mainly focus on targeting the EWS/FLI1 fusion protein, DNA damage repair pathways, tyrosine kinase inhibition, immunotherapy, and cell therapies [46]. Although multiple randomized trials are being conducted involving many international groups, the outcomes have been disappointing, indicating the need for novel therapies [47]. We previously used a multi-pronged approach using in silico predictions of drug activity via an integrated bioinformatics approach in parallel with an in vitro screen of FDA-approved drugs and identified key molecules, some of which include mitotic proteins that are essential for EWS progression [3]. Aberrant cell cycle progression is the key hallmark for cancers, and a variety of cell cycle agents have been used for the treatment of cancers in the clinic, including EWS [48].

Among these cell cycle inhibitors, mitotic inhibitors specifically prevent the formation of bipolar spindles and the normal assembly of chromosomes, leading to mitotic arrest and apoptosis [49]. Mitotic inhibitors have been used in clinical trials for several different cancer types, but major limitations associated with the use of these inhibitors include off-target toxicity affecting non-neoplastic cells and tumor recurrence associated with monotherapies [50]. Mitotic inhibitors can cause mitotic slippage, leading to aneuploidy and chromosomal instability and causing drug resistance [51]. Therefore, it is of critical importance to a use combination of drugs that target multiple pathways and synergistically inhibit tumor growth. Synergistic drug combinations have greater potency at lower and physiologically relevant doses compared to monotherapies. In this study, we employed a synergistic combination of mitotic inhibitors targeting KIF11 and AURKA to halt EWS tumor growth in vitro and in vivo, as assessed in a mouse xenograft model.

KIF11 [52] and AURKA [12] are overexpressed in many different cancer types, indicating the importance of this mitotic machinery to facilitate aggressive tumor growth (Figure 1A). We leveraged publicly available datasets to assess the RNA expression of these proteins in EWS patient tumor samples and found uniform expression in patient tumors irrespective of the disease state, indicating the feasibility of targeting these proteins in EWS patients (Appendix A). We tested different KIF11 inhibitors, including ispinesib, filanesib, and SB-743921, in combination with VIC-1911 (Figure 2). Ispinesib, SB-743921, and filanesib were tested in multiple clinical trials for several malignancies alone and in combination with other chemotherapies [28]. Ispinesib (SB-715992) was tested in refractory solid tumors in a pediatric phase I clinical trial and was found to be well tolerated; however, primary dose-limiting adverse events such as neutropenia and hepatotoxicity were observed at 9 mg/m^2^/dose MTD [22]. This finding suggests that KIF11 inhibitors are highly clinically relevant, and since they have never been tested for EWS patients before, they have the potential to be developed as a new treatment option for these patients. SB-743921 was synthesized to be more selective and potent by replacing the quinazoline ring in ispinesib with a chromen-4-one ring, resulting in a five-fold increase in its potency over the parent compound [53]. SB-743921 was used for the remainder of the study, as we observed it to be very potent, and to have the highest synergy with VIC-1911 in comparison with other KIF11 inhibitors. AURKA inhibitors such as alisertib were tested in phase 2 pediatric clinical trials at 80 mg/m^2^/dose, and myelosuppression was the most frequent toxicity observed [54]. Overall, these preclinical studies that used KIF11i or AURKAi as monotherapies suggest a potential clinical benefit if tested in combination and at lower doses.

One of the main mechanisms of action mediated by the drug combination we found was by enhancing G2/M-mediated cell cycle arrest and subG1 cells (Figure 5), ultimately resulting in enhanced apoptosis-mediated cell death (Figure 6). Further studies need to be conducted to identify additional mechanisms of action and pathways affected by this combination. We also tested other cancer cell lines, such as osteosarcoma and rhabdomyosarcoma, and observed synergy with this drug combination in our preliminary studies (Appendix A), which is consistent with elevated KIF11 and KIF15 levels (Figure 1A), indicating the potential for this drug combination in targeting other tumor types, especially sarcomas. Finally, mouse xenograft studies showed that the combination treatment of SB-743921 and VIC-1911 led to a significant delay in tumor growth compared to the single drugs alone, even though the tumors eventually developed therapy resistance (Figure 7 and Appendix A). Although the experiments with the higher levels of SB-743921 (2.5 mg/kg) showed sustained tumor regression and increased overall survival in combination with VIC-1911 (Appendix A), we repeated the experiments with lower levels of SB-743921 (Figure 7) due to significant body weight loss in mice (Appendix A). Mechanisms that mediate therapeutic resistance to the combination drug treatment could be attributed to several factors, including the presence of refractory cell populations in the tumor such as cancer stem cells [55] or upregulation of drug efflux mechanisms in tumor cells [56]. These areas are actively studied when new drugs are introduced into the clinic. Nevertheless, together, these in vivo studies warrant additional studies to evaluate different dosing schemes, along with assessing other classes of KIF11 inhibitors in combination with VIC-1911, to provide support for potential clinical translation.

Systemic cytotoxicity is often one of the major limiting factors for many chemotherapeutic regimens; hence, developing an effective drug delivery approach will largely benefit the field of precision medicine in general and specifically the area of targeted therapy. Future studies will be focused on employing targeted drug delivery approaches given the effectiveness of the drug combination that we observed in the current study. Overall, we identified a novel combination of mitotic inhibitors targeting KIF11 and AURKA that is highly synergistic in inhibiting the growth of an aggressive tumor such as Ewing sarcoma. Our findings are highly relevant, timely, and clinically translatable given the lack of proper therapies for this rare and orphaned pediatric disease.

## 5. Conclusions

This study identified a synergistic combination of inhibitors for KIF11 (SB-743921) and AURKA (VIC-1911) that demonstrated significant pre-clinical activity in vitro and efficacy in an in vivo xenograft mouse model of Ewing sarcoma. We further demonstrated that the drug combination reduces colony-plating efficiency, promotes cell death via G2/M arrest, and promotes apoptosis. Further studies will be required to define whether these are the best-in-class drugs to use and the dosing schedule to administer in order to minimize drug toxicity. These drugs have been tested in early phase clinical trials either individually or in combination with other drugs for several adult and pediatric tumor types and have shown acceptable safety profiles. Our studies support considering dual targeting of these biological pathways for future clinical trial design to address an unmet need in children and young adults diagnosed with EWS.

## Figures and Tables

**Figure 1 cancers-15-04911-f001:**
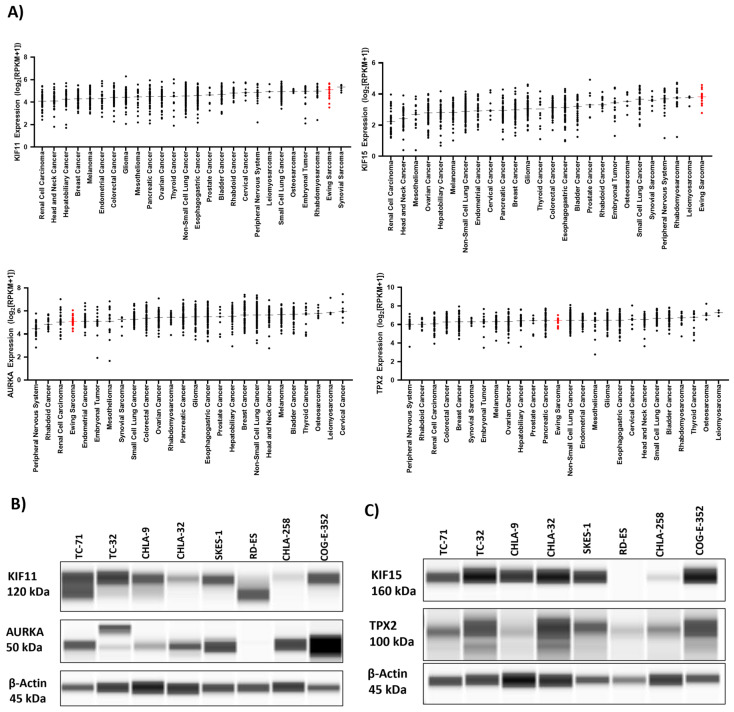
An in silico bioinformatics screen identifies mitotic proteins essential for EWS growth. (**A**) The DepMap portal was used to access the RNA expression data across different cancer cell lines. Expression in Ewing sarcoma is highlighted in red. Capillary-based analysis of protein lysates from EWS cell lines indicating expression of (**B**) KIF11 and AURKA and (**C**) KIF15 and TPX2 protein levels. The uncropped blots are shown in Appendix A.

**Figure 2 cancers-15-04911-f002:**
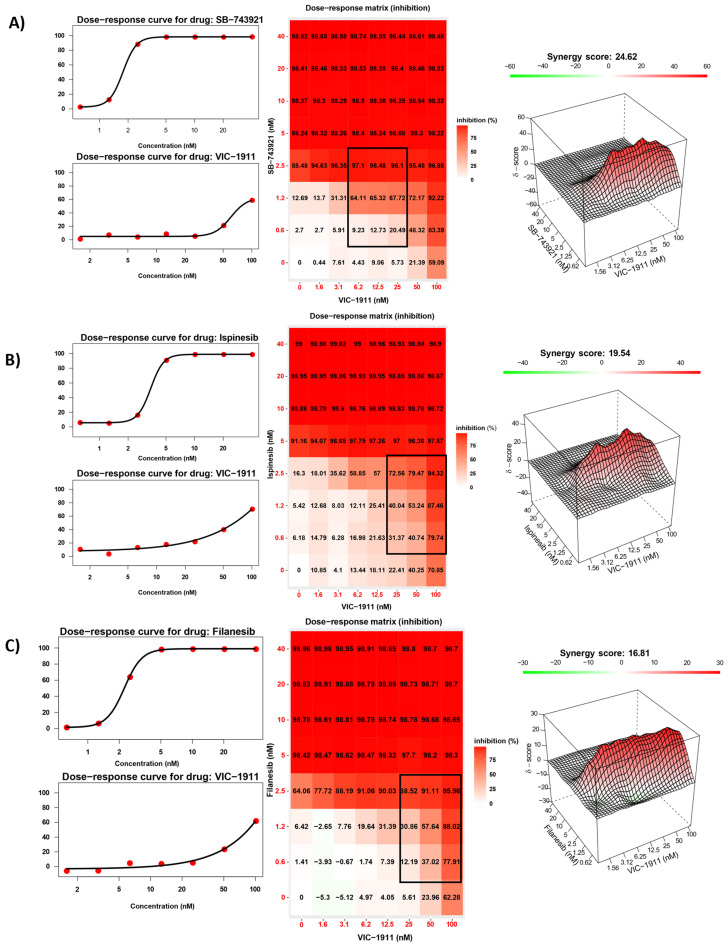
Synergistic inhibition of EWS growth by VIC-1911 and different KIF11 inhibitors. Dose–response curves, dose–response matrix, and heat map indicating synergy scores in TC-71 EWS cell line. (**A**) SB-743921 and VIC-1911 combination resulted in a Bliss synergy score of 24.62 for the highlighted drug combination in black, (**B**) the ispinesib and VIC-1911 combination resulted in a Bliss synergy score of 19.54 for the highlighted drug combination in black, and (**C**) the filanesib and VIC-1911 combination resulted in a Bliss synergy score of 16.81 for the highlighted drug combination in black. Biological triplicates (mean ± SEM, n = 3) were used for all drug combinations tested in the synergy assays.

**Figure 3 cancers-15-04911-f003:**
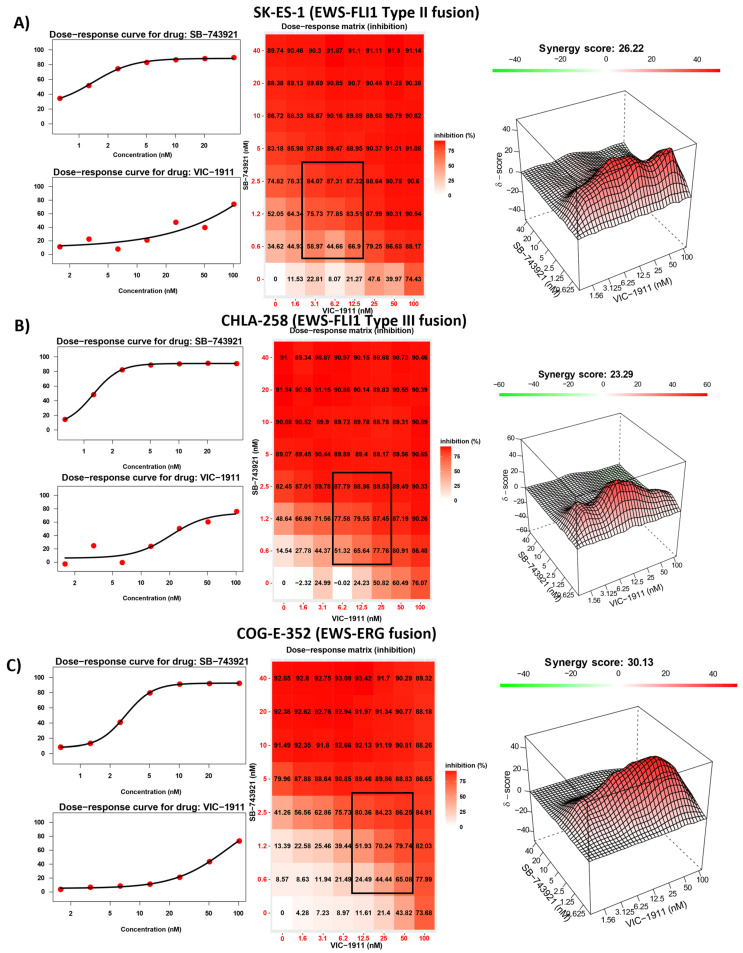
Synergy is observed in different EWS-ETS fusion-bearing cell lines. Dose–response curves, dose–response matrix, and heat map indicating synergy scores in different EWS cell lines. Combination treatment with SB-743921 and VIC-1911 resulted in following synergy scores for doses highlighted in black on the dose-response matrix heatmap (**A**) SK-ES-1 (26.22), (**B**) CHLA-258 (23.29), and (**C**) COG-E-352 (30.13). Biological triplicates (mean ± SEM, n = 3) were used for all drug combinations tested in the synergy assays.

**Figure 4 cancers-15-04911-f004:**
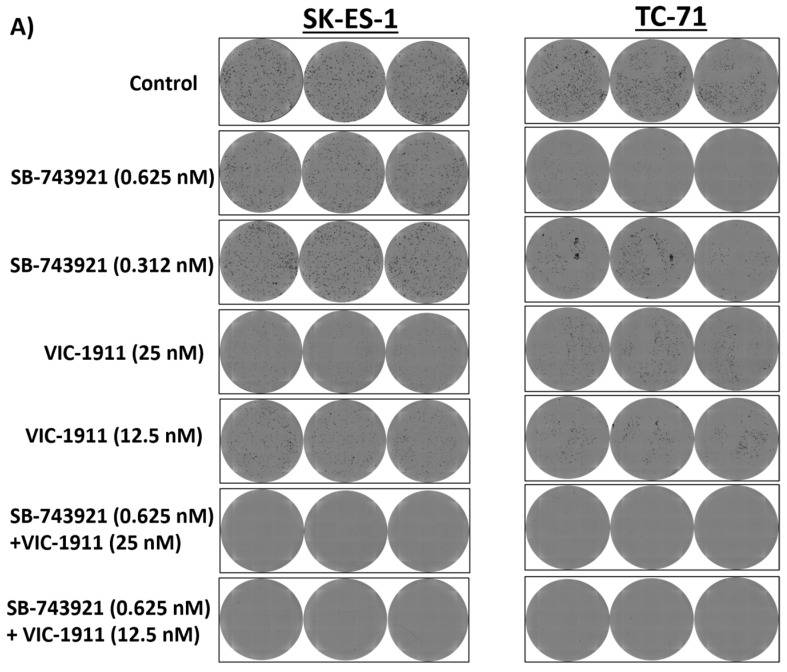
Colony formation assay upon combination treatment indicates reduced tumorigenicity. SK-ES-1 and TC-71 cells were treated with single drugs and combination and were incubated for one week post-treatment. The colonies formed were stained with crystal violet and images were taken by a Celigo Imager. (**A**) Images of triplicates per each condition are represented for control, SB-743921 (0.625 nM), SB-743921 (0.312 nM), VIC-1911 (25 nM), VIC-1911 (12.5 nM), and combination groups SB-743921 (0.625 nM) +VIC-1911 (25 nM) and SB-743921 (0.625 nM) + VIC-1911 (12.5 nM). (**B**) Plating efficiency of cells is represented; * *p* ≤ 0.05, ** *p* ≤ 0.01, *** *p* ≤ 0.001, **** *p*-value ≤ 0.0001, as assessed by one-way ANOVA.

**Figure 5 cancers-15-04911-f005:**
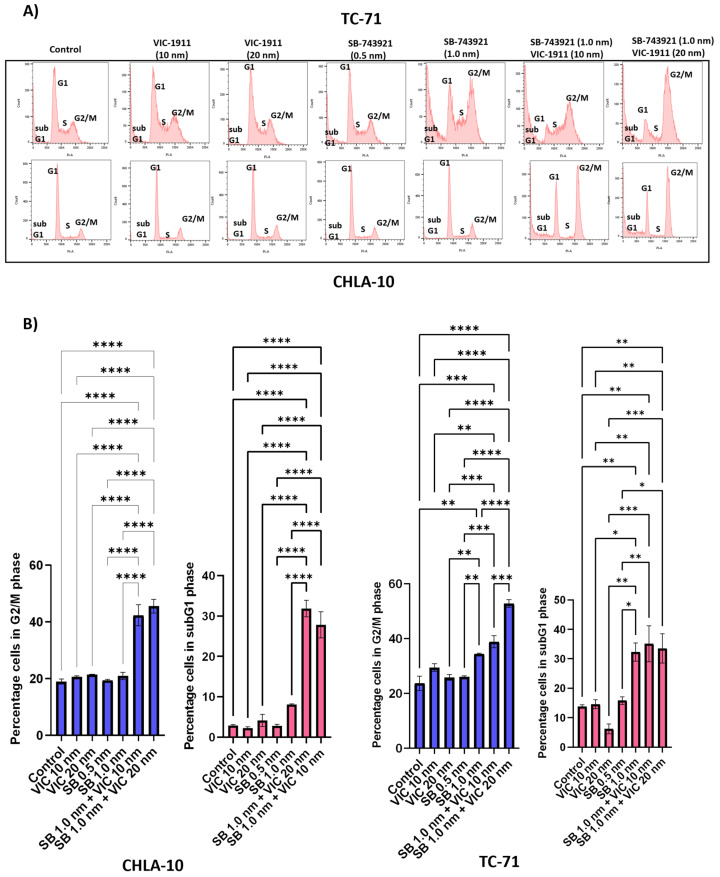
Combination treatment enhances G2/M cell cycle arrest of EWS cells. Cell cycle analysis was performed on CHLA-10 and TC-71 cells upon drug treatments and cells were assessed for changes in cell cycle profile 24 h post treatment after propidium iodide staining. Different phases of cell cycles are represented for different groups, including (**A**) control, VIC-1911 (10 nM), VIC-1911 (20 nM), SB-743921 (0.5 nM), SB-743921 (1.0 nM), and combination treatment (SB-743921 1.0 nM + VIC-1911 10 nM and SB-743921 1.0 nM + VIC-1911 20 nM). (**B**) Percentage of cells in G2/M phase and subG1 phase of cell cycle is represented by the bar graphs; * *p* ≤ 0.05, ** *p* ≤ 0.01, *** *p* ≤ 0.001 and **** *p*-value ≤ 0.0001, as assessed by one-way ANOVA.

**Figure 6 cancers-15-04911-f006:**
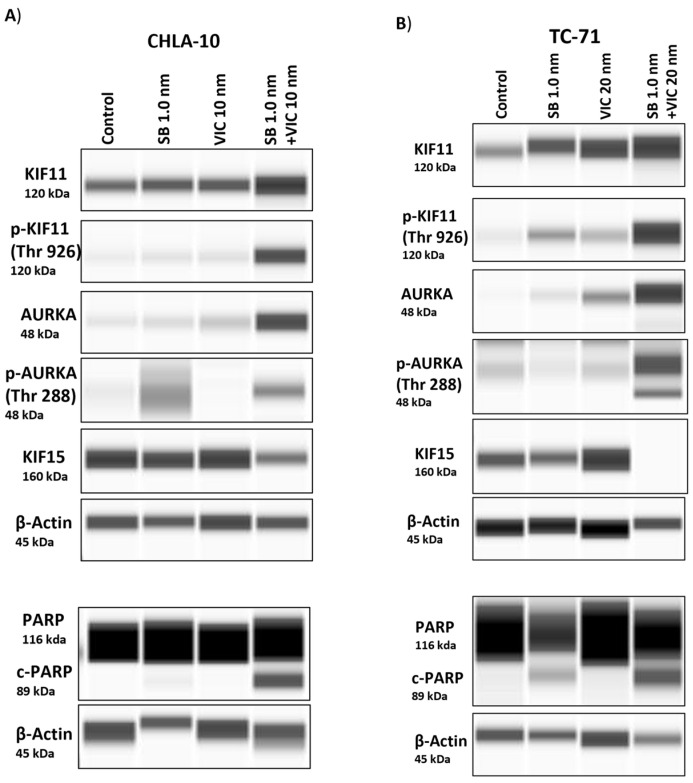
Analysis of protein expression post-drug treatment. (**A**) CHLA-10 and (**B**) TC-71 cells treated with drugs were assessed for changes in protein expression 24 h post-treatment via capillary electrophoresis-based Wes analysis. Increased protein levels of KIF11, p-KIF11^Thr926^ AURKA, and p-AURKA^Thr288^ were observed for the drug combination group, whereas KIF15 levels were noticeably lower. Similarly, enhanced cleaved-PARP expression was observed with the combination treatment. The uncropped blots are shown in Appendix A.

**Figure 7 cancers-15-04911-f007:**
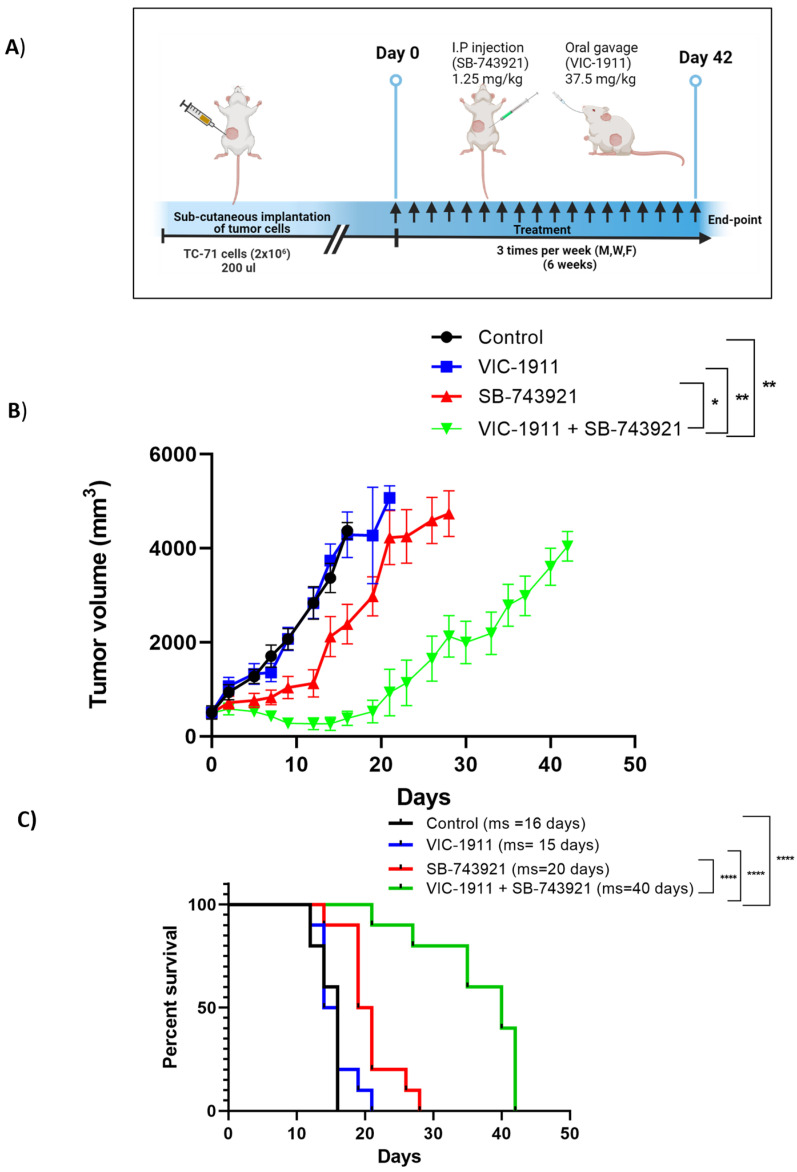
Tumor efficacy studies in a TC-71 xenograft model. (**A**) Timeline of in vivo study is represented. SB-743921 (1.25 mg/kg) and VIC-1911 (37.5 mg/kg) were dosed intraperitoneally and orally, respectively. Mice were treated with these drugs 3 times a week for 6 weeks (18 treatments) until end-point was reached. We observed significant (**B**) tumor regression (control vs. combination, SB-743921 vs. combination and VIC-1911 vs. combination; ** *p* ≤ 0.01, * *p* ≤ 0.05, as assessed by one-way ANOVA) at day 28. (**C**) Kaplan–Meier curves indicate overall survival with the combination treatment (VIC-1911 and SB-743921) compared to monotherapy and vehicle control groups (control vs. combination (**** *p* ≤ 0.0001), VIC-1911 only vs. combination (**** *p* ≤ 0.0001), and SB-743921 only vs. combination (**** *p* ≤ 0.0001), as measured by the log-rank (Mantel–Cox) test. Median survival for each group is represented on the survival curve.

**Table 1 cancers-15-04911-t001:** List of top 15 genes identified as essential for EWS tumor progression.

Gene Rank	Gene ID	Gene Symbol	Name	Pathway
**1**	**56992**	**KIF15**	**Kinesin family member 15**	**Motor proteins**
2	1058	CENPA	Centromere protein A	Mitosis, chromosome segregation, and cytokinesis
3	7153	TOP2A	DNA topoisomerase II alpha	Platinum drug resistance
4	5502	PPP1R1A	Protein phosphatase 1, regulatory (inhibitor) subunit 1A	Adrenergic signaling
5	51361	HOOK1	Hook microtubule-tethering protein 1	Vesicle trafficking
**6**	**6790**	**AURKA**	**Aurora kinase A**	**Oocyte meiosis**
7	5733	PTGER3	Prostaglandin E receptor 3 (subtype EP3)	Calcium signaling
8	2619	GAS1	Growth arrest-specific 1	Membrane trafficking
9	23306	TMEM194A	Transmembrane protein 194A	Nuclear envelope stiffness
10	1612	DAPK1	Death-associated protein kinase 1	Autophagy
11	29028	ATAD2	ATPase family, AAA domain-containing 2	Transcriptional activator
12	783	CACNB2	Calcium channel, voltage dependent beta 2 subunit	MAPK-signaling pathway
13	9787	DLGAP5	Discs, large (Drosophila) homolog-associated protein 5	Centrosome and spindle formation
**14**	**22974**	**TPX2**	**TPX2, microtubule-associated homolog (Xenopus laevis)**	**Regulation of kinetochore-microtubule interactions**
15	2956	MSH6	Muts homolog 6	Mismatch repair

## Data Availability

Data are contained within the article or Appendix A.

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
