# Peer review of "Inducing Mitotic Catastrophe as a Therapeutic Approach to Improve Outcomes in Ewing Sarcoma"

_cancers, 2023, doi:10.3390/cancers15204911_

Round 1

Reviewer 1 Report

Thanks for the opportunity to review this interesting paper on Ewing sarcoma.

Patients affected by these tumors have unfavorable prognoses despite the intensive therapy regimens used. Moreover, the conventional chemotherapy employed in the treatment of Ewing sarcoma often results in systemic toxicity. As a result, there is a necessity for further investigation into innovative therapeutic approaches that can enhance outcomes while decreasing the toxicities associated with treatment for these individuals. This study explored a potential treatment using mitotic inhibitors targeting KIF11 and AURKA in both in vivo and in vitro models. If the clinical trials confirm the efficacy of these inhibitors, they could serve as promising treatment options for Ewing sarcoma in the future.

The research examines the efficacy of mitotic inhibitors that target KIF11 and AURKA in inhibiting both in vivo and in vitro models of Ewing sarcoma. The use of xenograft tumors in mice adds an intriguing aspect to the study. It provides valuable information on how these inhibitors affect tumor volume and overall survival, although histological analyses were not included as part of the experimental design. Histological analysis is an essential part of preclinical studies using xenograft models. It provides valuable insights into the effects of treatment on tumors, including cellular composition, necrosis, apoptosis, proliferation changes, and alterations in the tumor microenvironment. Without histological examinations, it is difficult to understand the biological changes caused by treatment within tumors accurately. To enhance the comprehensiveness and scientific rigor of future iterations of this study, incorporating histological evaluations would be beneficial. This addition will strengthen both the scientific validity and translational impact of the research by providing a more comprehensive characterization of treatment effects.

The paper extensively employs various methods to assess the impact of the drug combination on Ewing sarcoma cell lines in vitro. These experiments provide valuable insights into immediate cellular responses, including cell-cycle arrest and cell death. However, additional molecular studies would be necessary to gain a more comprehensive understanding of the underlying mechanisms behind these drug actions.

Incorporating various techniques such as gene expression profiling, proteomic analysis, and specialized assays that focus on important signaling molecules can enhance the quality of this research. Examining the molecular pathways involved in drug action on cells, after the cells are treated, will provide valuable insights into specific targets and biomarkers associated with their effectiveness.

One notable aspect of the paper is the absence of a direct comparative study on the individual effects and mechanisms of action of AURKA and KIF11 in vitro. While the combined drug regimen effectively inhibits Ewing sarcoma cell viability and induces cell-cycle arrest, it would have been informative to include separate investigations on each drug alone. Performing comparative studies on these distinct drugs could have provided valuable insights into their independent effectiveness and mechanisms of action. 

In conclusion, this study explores the efficacy of a drug combination targeting KIF11 and AURKA in inhibiting Ewing sarcoma tumor growth. The results provide valuable insights into the impact on tumor volume and overall survival. However, additional studies may be necessary to better understand why this drug experiment may be more effective compared to other combinations and uncover its underlying molecular mechanisms.

Author Response

Reviewer 1: “Thanks for the opportunity to review this interesting paper on Ewing sarcoma”.

R1.1. “Patients affected by these tumors have unfavorable prognoses despite the intensive therapy regimens used. Moreover, the conventional chemotherapy employed in the treatment of Ewing sarcoma often results in systemic toxicity. As a result, there is a necessity for further investigation into innovative therapeutic approaches that can enhance outcomes while decreasing the toxicities associated with treatment for these individuals. This study explored a potential treatment using mitotic inhibitors targeting KIF11 and AURKA in both in vivo and in vitro models. If the clinical trials confirm the efficacy of these inhibitors, they could serve as promising treatment options for Ewing sarcoma in the future”.

Response: We thank the reviewer for their encouraging comments on the manuscript.

R1.2. “The research examines the efficacy of mitotic inhibitors that target KIF11 and AURKA in inhibiting both in vivo and in vitro models of Ewing sarcoma. The use of xenograft tumors in mice adds an intriguing aspect to the study. It provides valuable information on how these inhibitors affect tumor volume and overall survival, although histological analyses were not included as part of the experimental design. Histological analysis is an essential part of preclinical studies using xenograft models. It provides valuable insights into the effects of treatment on tumors, including cellular composition, necrosis, apoptosis, proliferation changes, and alterations in the tumor microenvironment. Without histological examinations, it is difficult to understand the biological changes caused by treatment within tumors accurately. To enhance the comprehensiveness and scientific rigor of future iterations of this study, incorporating histological evaluations would be beneficial. This addition will strengthen both the scientific validity and translational impact of the research by providing a more comprehensive characterization of treatment effects”.

Response: Our initial studies only focused on measurement of tumor growth and survival. Future studies will include histology and tumor characterization studies. Given, the short time frame for resubmission (1 week), we are unable to perform any additional animal studies.

R1.3. “The paper extensively employs various methods to assess the impact of the drug combination on Ewing sarcoma cell lines in vitro. These experiments provide valuable insights into immediate cellular responses, including cell-cycle arrest and cell death. However, additional molecular studies would be necessary to gain a more comprehensive understanding of the underlying mechanisms behind these drug actions. Incorporating various techniques such as gene expression profiling, proteomic analysis, and specialized assays that focus on important signaling molecules can enhance the quality of this research. Examining the molecular pathways involved in drug action on cells, after the cells are treated, will provide valuable insights into specific targets and biomarkers associated with their effectiveness”.

Response: We thank the reviewer for their suggestion. We are currently working on investigating the mechanism of action of this drug combination as a follow up study to this manuscript.

R1.4. “One notable aspect of the paper is the absence of a direct comparative study on the individual effects and mechanisms of action of AURKA and KIF11 in vitro. While the combined drug regimen effectively inhibits Ewing sarcoma cell viability and induces cell-cycle arrest, it would have been informative to include separate investigations on each drug alone. Performing comparative studies on these distinct drugs could have provided valuable insights into their independent effectiveness and mechanisms of action”.

Response: We would like to clarify that in all experiments performed with the drug combination, we included single drug only controls for comparisons.

R1.5. “In conclusion, this study explores the efficacy of a drug combination targeting KIF11 and AURKA in inhibiting Ewing sarcoma tumor growth. The results provide valuable insights into the impact on tumor volume and overall survival. However, additional studies may be necessary to better understand why this drug experiment may be more effective compared to other combinations and uncover its underlying molecular mechanisms”.

Response: We agree with the reviewer, and as such additional mechanistic studies are underway and are beyond the scope of this initial manuscript.

Reviewer 2 Report

This study identified a synergistic combination of inhibitors for KIF11 and AURKA that demonstrate significant pre-clinical activity in vitro and in vivo in Ewing sarcoma. The results are very interesting for future clinical assays to try to find new therapeutic targets. I consider these preclinical studies are important to find new treatments.

Author Response

Reviewer 2: “This study identified a synergistic combination of inhibitors for KIF11 and AURKA that demonstrate significant pre-clinical activity in vitro and in vivo in Ewing sarcoma. The results are very interesting for future clinical assays to try to find new therapeutic targets. I consider these preclinical studies are important to find new treatments”.

Response: We thank the reviewer for their positive comments about the manuscript.

Reviewer 3 Report

Conclusions must be used the results. The last sentences is not real that the paper showed the results in mice model only. That the next study according the good clinical practice must be use for the conclusions the clinical application in childern about. 

Author Response

Reviewer 3: “Conclusions must be used the results. The last sentences is not real that the paper showed the results in mice model only. That the next study according to the good clinical practice must be use for the conclusions the clinical application in children about”.

Response: We have revised the conclusions section of the manuscript and included more details of the results of our study in pre-clinical in vitro and in vivo mouse models of Ewing Sarcoma.

Reviewer 4 Report

In the manuscript by Turage et al., the authors describe a combined approach to induce mitotic catastrophe in Ewing sarcoma cell lines in vitro and in vivo using xenograft models in mice. The study is well-designed, and the selection of drug combinations and establishment of synergism was carefully done and presented. The results are convincing and the explanation of the presented results is clear. The conclusions are mostly well substantiated. 

A few minor comments have to be addressed:

The overall expression of KIF11, AURKA, KIF15 and TPX2 in various cancer types is well presented. It would be interesting to show, how the expression of these genes looks in different normal tissues, for example using GTEX data or other sources. This might be helpful to identify potentially affected normal cell pools in the body, presumably highly proliferating tissues. 

This brings us to the general comment, where some of the conclusions are slightly overstated, there must be information about the severity of the clinical trials and would have been interesting to see the effect in normal tissue-derived cells either with similar duplication time as in EWS cell lines or having EWS cells in slightly staved conditions to reduce proliferation rates closer to ones in vivo to mimic the mitigated effect in the less fast proliferating cell. 

This, in part, might explain the observed, resistance in the xenograft model. This observation should be strongly noted and explained in the manuscript as (extent the lines that started in lines 503-506)  it may show the inevitable resistance that may occur when the treatment is applied in patients, perhaps with even larger tumour volume to start with. 

Line 165: "(1,50,000 per well) needs correction for sure for the proper number and format.

Author Response

Reviewer 4: “In the manuscript by Turaga et al., the authors describe a combined approach to induce mitotic catastrophe in Ewing sarcoma cell lines in vitro and in vivo using xenograft models in mice. The study is well-designed, and the selection of drug combinations and establishment of synergism was carefully done and presented. The results are convincing, and the explanation of the presented results is clear. The conclusions are mostly well substantiated”.

Response: We thank the reviewer for their positive comments.

“A few minor comments have to be addressed”:

 R4.1. “The overall expression of KIF11, AURKA, KIF15 and TPX2 in various cancer types is well presented. It would be interesting to show, how the expression of these genes looks in different normal tissues, for example using GTEX data or other sources. This might be helpful to identify potentially affected normal cell pools in the body, presumably highly proliferating tissues”.

Response: We thank the reviewer for their suggestion, and have included the GTEx data for KIF11, AURKA, KIF15 and TPX2 in the Supplementary Figure 4.

R4.2. “This brings us to the general comment, where some of the conclusions are slightly overstated, there must be information about the severity of the clinical trials”.

ResponseWe have included some information on clinical trials with KIF11 and AURKA inhibitors in pediatric population in the discussion section (lines 488-491 & 498-502).

R4.3. “Would have been interesting to see the effect in normal tissue-derived cells either with similar duplication time as in EWS cell lines or having EWS cells in slightly staved conditions to reduce proliferation rates closer to ones in vivo to mimic the mitigated effect in the less fast proliferating cell. This, in part, might explain the observed resistance in the xenograft model. This observation should be strongly noted and explained in the manuscript as (extent the lines that started in lines 503-506) it may show the inevitable resistance that may occur when the treatment is applied in patients, perhaps with even larger tumor volume to start with”.

Response: We have included the synergy data for mesenchymal stem cells in the Supplementary Figure 6C. The doubling of MSCs is approximately 26-28 hours and is like most of the EWS cell lines we have used in the manuscript (TC-71 is 20-24 hours, CHLA-10 – 24-32 hours). Despite having similar doubling time, the drug combination has high synergy in cancer cells but not in normal cells. We also revised and added more information (lines 518- 521) in the results section as per reviewers’ suggestion.

R4.4. “Line 165: "(1,50,000 per well) needs correction for sure for the proper number and format”.

Response: We have revised the cell number to 1.5 x105 cells.

Round 2

Reviewer 1 Report

Thank you for your feedback. I am pleased with your responses. I look forward to future in vivo experiments incorporating histology techniques. Furthermore, I hope your future molecular studies will elucidate the underlying molecular mechanisms.